# Pathfinder-Driven Chemical Space Exploration and Multiparameter Optimization in Tandem with Glide/IFD and QSAR-Based Active Learning Approach to Prioritize Design Ideas for FEP+ Calculations of SARS-CoV-2 PL^pro^ Inhibitors

**DOI:** 10.3390/molecules27238569

**Published:** 2022-12-05

**Authors:** Njabulo Joyfull Gumede

**Affiliations:** Department of Chemistry, Mangosuthu University of Technology, P.O. Box 12363, Jacobs 4026, South Africa; ngumede@mut.ac.za; Tel.: +27-31-907-9396

**Keywords:** Pathfinder, reaction-based enumeration, active learning QSAR, FEP+, SARS-CoV-2 papain-like protease

## Abstract

A global pandemic caused by the SARS-CoV-2 virus that started in 2020 and has wreaked havoc on humanity still ravages up until now. As a result, the negative impact of travel restrictions and lockdowns has underscored the importance of our preparedness for future pandemics. The main thrust of this work was based on addressing this need by traversing chemical space to design inhibitors that target the SARS-CoV-2 papain-like protease (PL^pro^). Pathfinder-based retrosynthesis analysis was used to generate analogs of GRL-0617 using commercially available building blocks by replacing the naphthalene moiety. A total of 10 models were built using active learning QSAR, which achieved good statistical results such as an R^2^ > 0.70, Q^2^ > 0.64, STD Dev < 0.30, and RMSE < 0.31, on average for all models. A total of 35 ideas were further prioritized for FEP+ calculations. The FEP+ results revealed that compound **45** was the most active compound in this series with a ΔG of −7.28 ± 0.96 kcal/mol. Compound **5** exhibited a ΔG of −6.78 ± 1.30 kcal/mol. The inactive compounds in this series were compound **91** and compound **23** with a ΔG of −5.74 ± 1.06 and −3.11 ± 1.45 kcal/mol. The combined strategy employed here is envisaged to be of great utility in multiparameter lead optimization efforts, to traverse chemical space, maintaining and/or improving the potency as well as the property space of synthetically aware design ideas.

## 1. Introduction

A global pandemic, caused by what was initially referred to as the 2019 nCov virus, which started in 2019 in China and still ravages now, has wreaked havoc on humanity [1,2,3]. It was hypothesized that this virus is transmitted between humans through sneezing, coughing, and small saliva droplets released when an infected person speaks [4]. As a result, scientists were able to pinpoint the new virus as a member of the coronavirus family [5]. Later, a change in taxonomy was made from the 2019 nCoV, to the SARS-CoV-2 virus by the Corovidae study group of the International Committee on Taxonomy of Viruses [6]. The change was made since this new virus was derived from a similar type of virus, the severe acute respiratory syndrome (SARS-CoV), that led to a pandemic between 2002 and 2003 [1,6,7].

Coronaviruses were first discovered in the 1960s and were thought to be a group of related viruses causing common colds [4]. These common colds were found to be caused by viral strains such as HCoV 229E and HCoV OC43 [8]. The groups of these viruses are classified as α-, β-, γ-, and δ-CoVs from the point they were discovered in the 1960s [1,9]. Coronaviruses are single-stranded RNA-enveloped viruses that infect the host leading to respiratory, gastrointestinal, and neurological diseases [1,3,8,10]. CoVs throughout history have been transmitted from animals to humans through a zoonotic transmission event [1,11]. SARS-CoV-2 is different from the other related viruses in its class in that it is highly infectious, and the effect of a secondary attack is just as severe as the primary attack [12]. The main protease (MP^pro^) and papain-like protease (PL^pro^) are validated drug targets for the SARS-CoV-2 virus [13,14]. Inhibitors of the former work by blocking viral infection, while inhibitors of the latter block viral replication. Furthermore, PL^pro^ is also important for innate immunity [13,15]. As such, PL^pro^ is a clinically relevant drug target for its multiple roles, as it overcomes innate immunity by virtue of reversing processes related to ubiquitination and ISGylation [7,13,16]. In fact, for the successful infection of the host by the virus to occur, the cleavage of host proteins responsible for ubiquitination and ISGylation needs to occur, which is facilitated by PL^pro^ enzyme activities [17,18].

Efforts have been made in the past to use molecular modeling to design PL^pro^ inhibitors for SARS-CoV [19]. Ghosh et al. [20] and Ratia et al. [21] performed high-throughput screening assays in independent studies to screen commercial libraries of diverse drug-like molecules. The experimental results revealed that a non-covalent inhibitor GRL-0617, an analog of compound 7724772 (see Figure 1a), shows the highest level of potency. The lead compound 7724772 inhibits SARS-CoV PL^pro^ with an IC_50_ of 20 µM, while the optimized analog GRL-0617 (see Figure 1b) shows an IC_50_ of 0,60 µM. Further studies revealed that GRL-0617 inhibits Vero E6 cell lines for SARS-CoV viral replication with an EC_50_ of 15 µM, without any form of cytotoxicity demonstrated [18,19]. Recent studies by Freitas et al. [7] have also revealed that GRL-0617 is more potent than an analog, which is compound **6** from their study, and inhibits SARS-CoV-2 PL^pro^ with an IC_50_ of 2.4 µM, whilst compound **6** inhibits SARS-CoV-2 PL^pro^ with an IC_50_ of 5.0 µM. Attempts have recently been made to use Artificial Intelligence/Machine Learning (AI/ML) tools by several research groups to design SARS-CoV-2 drug-like molecules [21,22,23]. In fact, ML tools such as deep learning have transformed drug discovery and development. The ML models are trained using molecules with known activities to the target of interest [21,23,24]. Arshia et al. [24] have recently employed a combined strategy of the De Novo based design of SARS-CoV-2 M^pro^ inhibitors employing deep learning, docking, and MD simulations successfully. Furthermore, Murugesan et al. [25] and Patel et al. [26], in independent studies, have successfully screened phytochemical compounds derived from medicinal plants and used docking and MD simulations to predict their binding affinities when bound to SARS-CoV-2 M^pro^. With this in mind, the main aim of this study is to demonstrate the synergy between active learning-based Glide docking tools in tandem with Pathfinder reaction-based enumeration tools and physics-based relative binding affinity estimation tools. This study aims to optimize GRL-0617, a lead compound targeting SARS-CoV-2 PL^pro^, by traversing chemical space, designing its analogs, and predicting their binding affinities using FEP+. The design ideas that are predicted to be the most active will be selected and prioritized for synthesis. Further biological activities of the design ideas will be performed in the future and will be relevant as therapeutic interventions for our preparedness for future pandemics.

## 2. Materials and Methods

### 2.1. Computational Details

Schrödinger Life-Sciences Suite 2021-2 was utilized for all computational calculations. Maestro (v12.8) was used as a Graphical User Interface comprising various, diverse Schrödinger modules. Pathfinder, LigPrep, Protein Preparation Wizard, Prime, Glide, Induced Fit Docking, Desmond for Molecular Dynamics Simulation, Maximum Common Substructural (MCS) Docking, AutoQSAR, Force-field builder, and FEP+ were used.

### 2.2. Reaction-Based Enumeration Using Pathfinder

The 2-D sketcher panel on Maestro was used to draw the structure of GRL-0617, a lead compound with activity against the SARS-CoV-2 PL^pro^ drug target. Pathfinder-driven enumeration was performed on GRL-0617 with a maximum depth of 1. A total of 4 pathways were generated from a commercial library of building blocks via the retrosynthesis of known coupling reactions as follows: **Pathway 1**—amide_coupling_1, **Pathway 2**—amide_coupling_2, **Pathway 3**—curtius_3, and **Pathway 4**—Suzuki_2 (see Appendix A). In the first round of enumeration using **Pathway 1**, reactant 1 was retained as the original reactant and reactant 2 was selected from the reactant library of N containing heterocycles. The rest of the enumerations were composed of hydrazine aryl, primary and secondary amines, acid chloride, carboxylates, isocyanates, aryl, and vinyl halides. In all the enumeration runs, Smiles Arbitrary Target Specification (SMARTS) was used to remove ligands with reactive functional groups and Pan assay Interfering Structures (PAINS) offenders affecting the design ideas were filtered. A physicochemical filtering criterion was also used to filter design ideas that fall outside of the desired drug-like chemical space. Such as design ideas with an MW between 150 and 500, LogP between −1.50 and 5.0, TPSA between 30 and 150, HBA between 0 and 12, and HBD between 0 and 5, and a maximum number of rotatable bonds less than 10 was retained.

### 2.3. Ligand and Protein Preparation

#### 2.3.1. Ligand Preparation

A total of 89,529 enumerated design ideas were divided according to their enumeration circles from 1 to 10, as explained in the previous section. The 3-D coordinates of the design ideas were prepared by subjecting them to Ligprep [27]. We employed Epik [28,29,30] to generate possible ionization states, tautomers, stereoisomers, and conformers at pH 7.4. The OPLS4 forcefield was used for the restrained minimization of the resulting conformers.

#### 2.3.2. Protein Preparation

The X-ray crystal structure of SARS-CoV-2 PL^pro^ in complex with GRL-0617 (PDB ID: 7JIR) with a resolution of 2.09 Å was uploaded from RCSB Protein Data Bank (PDB) [31]. Bond orders were assigned, creating zero bonds to metals, creating disulfide bonds, and filling in missing side chains and loops using Prime, [32,33,34] using the default parameters. The protonation states of PL^pro^ and GRL-0617 were simulated at pH 7.4 using Epik [28,29,30]. Hydrogen bond assignment was optimized to sample water orientations, using crystal symmetry, and minimizing hydrogens of altered species. An interactive optimizer was performed at pH 7.0 using PROPKA [29] to minimize steric hindrance. Waters within 3 Å of the co-crystallized ligand in the active site were removed. Restrained minimization was undertaken by using a customized version of OPLS4 force field [27,28,29,30,31,32,33,34,35,36,37] to converge heavy atoms at a Root-Mean-Square Deviation (RMSD) of 0.30 Å.

### 2.4. Glide SP Docking and Auto QSAR Active Learning Models

The prepared structure of PL^pro^ described in Section 2.3.2 was used as an initial structure for receptor grid generation. The co-crystalized structure of GRL-0617 was used to identify the active site of PL^pro^ for positional constraints. The hydrogen bond constraints were employed by selecting all hydrogen bonds that the ligand makes with the amino acids of PL^pro^. Rotatable groups of the amino acids of PL^pro^ were selected including the excluded regions in the receptor that the ligand could not occupy. Glide docking [38], employing the SP mode, was used to place the enumerated compounds in the active site cavity of PL^pro^. Furthermore, Hydrogen bond constraints from the receptor grid were selected for docking. Finally, all default parameters for Glide docking were used as they were. A total of 10 rounds of Glide SP docking were performed on enumerated compounds to filter compounds that do not bind with PL^pro^ and those that exhibited weak binding when predicted with the Auto QSAR models [39]. The Auto QSAR models using the enumerated dataset were trained, tested, and re-trained using the enumerated dataset. This process continued until some sort of convergence was attained after 10 rounds of active learning Glide SP Auto QSAR modeling.

### 2.5. Induced Fit Docking (IFD) to Screen the Library of Enumerated Design Ideas

Top-scoring design ideas that were screened with Glide SP were used as starting structures for IFD [40,41] calculations. The co-crystalized ligand was picked to mark the active site cavity of PL^pro^ using an enclosing box in the prepared structure of 7JIR. A standard protocol for docking was selected, to generate 20 poses. Hydrogen bond constraints were applied and were selected from those between GRL-0617 and amino acids such as Asp164, Tyr273, and Gln269 in the active site cavity of PL^pro^. Default parameters for ligand conformational sampling during the first round of Glide docking were employed in IFD. Glide redocking was performed using the XP mode in the IFD panel. Rank ordering of the top-scoring poses was performed by using the procedures that we developed in prior publications [42,43].

### 2.6. Molecular Dynamics Simulations

Molecular dynamics simulations (MDS) [44,45] were performed on 7JIR using the Desmond package on Maestro v2.8, by setting up the system first. A predefined SPC explicit solvent system was enclosed in an orthorhombic box shape. The solvation model was placed in an enclosing box with a diameter of 10 Å. The model system was then neutralized by adding a NaCl salt solution at a concentration of 0.15 M. The minimization of the system was undertaken by using OPLS4 force field. Once the model system was created the output file was uploaded into the MDS panel on Maestro. The MD simulation time was set at 100 ns and a simulation trajectory of 100 ps with a total of 1000 frames. The NPT ensemble class was selected at a temperature of 300 K and a pressure of 1.01 bar.

### 2.7. Maximum Common Substructural (MCS) Docking

MCS docking [46] was performed using the MCS docking panel on Maestro to align all the structures of design ideas prioritized for FEP+, using the last 2ns frame of 7JIR generated by MDS in Section 2.6 above. The structure of GRL-0617 was extracted from the 2ns MDS frame and used as a reference compound in MCS docking. The structures of the selected design ideas were extracted as input structures from the IFD pose with the highest docking score.

### 2.8. Free Energy Perturbation Plus (FEP+) to Predict the Relative Binding Affinities of the Design Ideas

Prior to FEP+ [47,48,49,50] calculation, the MCS poses from Section 2.7 above were used as starting structures for FEP+ calculations. FEP+ methodology was performed by first using the last frame of MDS for 7JIR as an input structure for the protein, where 7JIR was split into the protein, ligand, and water. Then the unliganded 7JIR was added to the project table where all ligands’ outputs from the MCS docking were placed. The Force Field builder panel was used to sample the force field parameters, such as torsions, for the enumerated design ideas that are absent in OPLS4 force field. Ideally, Force Field builder employs quantum mechanical simulations available in Jaguar to fit the torsion parameters simulated and is then compared to those obtained from OPLS4 force field [51,52], since newly designed chemical entities usually contain torsion parameters that are not specifically represented by parameters in forcefield databases [53]. A ligand health check in the FEP+ panel was performed to establish whether the ligands fit the criteria of having new torsional parameters explicit. Relative binding affinity calculations were performed by using an FEP+ panel on Maestro. FEP+ is based on the Replica Exchange with Solution Tempering (REST) using an MD algorithm that utilizes OPLS4 force field and is available on the Desmond package [51,52,53]. The FEP+ simulation parameters used included the NVT ensemble, a simulation time of 20ns, and a random seed of 2007. Several λ windows started with a default of 12 λ windows, followed by core hopping with 16 λ windows, and lastly charged hopping with 16 λ windows. A cycle closure technique was used to generate a perturbation map, which was applied to attain the convergence error estimates of relative binding affinities, at a simulation window of 20 ns. GRL-0617 was used as a reference compound with known experimental IC_50_ to PL^pro^. As such, FEP+ is a well-established method for reliably predicting the relative and/or absolute binding affinities of novel compounds. In the literature, various scientists have prospectively and retrospectively demonstrated the great utility of FEP+ in hit-to-lead identification and lead optimization efforts [51,52,53,54,55,56]. Furthermore, a detailed theory behind the FEP+ method and the detailed protocols can be obtained from the original publications that were also adopted in this work [47,48,49,50,53,55,56,57,58,59,60,61].

## 3. Results and Discussions

### 3.1. Pathfinder Reaction-Based Enumeration

In the Pathfinder reaction-based enumeration panel, the structure of GRL-0617 was uploaded to rapidly explore the chemical space of the library of building blocks. The aim was to explore the structure–activity relationship and potency of the design ideas. Furthermore, the multiparameter optimization of the lead compound is also aimed at improving and/or maintaining the physicochemical and Adsorption Distribution Metabolism and Excretion (ADME) properties important in drug discovery and development. Various design test cycles (Figure 2) were performed with the aim of yielding optimum design ideas, using the method adapted from Konze et al. [54] and Ghanakoda et al. [61]. All 10 rounds of ideation involved enumerating building blocks such as Nitrogen containing-heterocycles (1st round of enumeration), primary and secondary amines (2nd round of enumeration), aryl and vinyl halides (3rd round of enumeration), acid chlorides (4th round of enumeration), carboxylates (R^1^) and primary and secondary amines (R^2^) (5th round of enumeration), carboxylates only (7th round of enumeration), Isocyanates (8th round of enumeration), and carboxylates and trifluoroborates (9th and 10th round of enumerations). Appropriate physicochemical, SMARTS, and PAINS filters were employed on the Pathfinder panel (Appendix A).

### 3.2. Glide SP Docking and Filtering of Enumerated Design Ideas

Glide SP docking employing hydrogen bond and core constraint was used to place the design ideas in the active site cavity of a prepared grid file for PL^pro^. Different rounds of Glide SP docking calculations were performed, informed by the number of enumerations runs, which generated diverse ideas with the hope of expanding the chemical space of PL^pro^ inhibitors. Firstly, ligprep generated 30,514 output structures of Nitrogen-containing heterocycle design ideas. This was followed by filtering design ideas based on a proprietary set of SMARTS patterns and PAINS filters. This step was performed since the conversion of structures from 2D to 3D in some cases generates improper tautomers [60]. Design ideas with a formal charge of zero were retained, generating neutral drug-like design ideas including GRL-0617 and its 10 analogs with known experimental IC_50_ values from the literature for further Glide SP docking. The positive controls used were analogs of compound 7724772 (Figure 1a) and were designed together with GRL-0617 and tested for bioanalysis using the same assay. These were included because they already have the IC_50_ values, so they could train, test, re-train, and re-test the QSAR models. A total of 10 rounds of Glide SP docking facilitated by active learning QSAR models were performed. This was accomplished by training and testing QSAR model_1 up to QSAR model_10 to predict the docking scores of design ideas. The docking of Nitrogen-containing heterocycles generated 638 poses that fit the active site cavity of PL^pro^. The Glide SP docking of a hydrazine-aryl-containing subset generated a total of 997 poses, and 88 poses were selected. The Glide SP docking of an acid chlorides-containing subset generated a total of 996 poses, and only 65 poses were selected. Furthermore, the Glide SP docking of acid chlorides and primary and secondary amine ideas to PL^pro^ generated 582 poses, and 55 of those were selected. The Glide SP docking of aryl and vinyl halides generated 963 poses, and 222 of those were selected. Furthermore, the Glide SP docking of carboxylates and primary and secondary amine series generated 900 poses, and 92 of those were selected. The Glide SP docking of a carboxylate-only series generated 557 poses, of which 147 were selected. The Glide SP docking screening of design ideas containing isocyanate building blocks generated 146 top-scoring poses that were selected. These 10 rounds of screening the design ideas with Glide SP docking and predicting their docking scores using various Auto QSAR models prior to docking demonstrated the capabilities of the active learning approach coupled with docking to screen a large database of enumerated compounds against the PL^pro^ receptor (Figure 2).

### 3.3. Auto QSAR-Based Active Learning Models to Prioritize Ideas for Glide/IFD XP Docking

As previously discussed in Section 3.3 above, the Auto QSAR models were trained on Pathfinder-generated molecules that were filtered by Glide SP docking based on their ability to bind with PL^pro^. An active learning approach was adopted to develop models that were used to filter enumerated compounds and select compounds that would further be docked with a more in-depth scoring function such as IFD (Figure 2). The models were built to further assess their ability to predict the Glide SP docking scores (an independent variable) against physicochemical properties and topological descriptors (dependent variables) retrieved from Canvas in situ. Multivariate statistical methods such as Multiple Linear Regression (MLR), Partial Least Squares Regression (PLS), and Kernel-based PLS (KPLS) were employed on the enumerated compounds. The descriptors used include binary fingerprints i.e., radial, linear, dendritic, and 2-D molecular prints. These were used in combination with numeric descriptors such as topographical, physicochemical, and ligand filters. A numerical rather than a categorical model was applied to the Auto QSAR panel to generate and build the Auto QSAR models.

Table 1 details the models built using the subsets of enumerated design ideas, accompanied by their statistical parameters following an active learning approach. In this approach, all models reported in Table 1 were trained to learn to predict the docking scores of a diverse set of design ideas targeting SARS-CoV-2 PL^pro^. The models, Auto QSAR _model_1 up to Auto QSAR _model_10, were generated using KPLS, but they differed in terms of binary fingerprints, since the enumerated design ideas have different building blocks attached in place of the naphthalene moiety (Table 1 and Table 2). The most significant models with good R^2^ > 0.70, Q^2^ > 0.64, STDDEV < 0.30, and RMSE < 0.31 were models **6**, **7**, **8,** and **10** (Table 1). Hence, models **8** and **10** were selected to predict the binding affinities of the compounds that remained and did not undergo Glide SP docking before all the QSAR models were built. Auto QSAR _model_1 was trained with design ideas exhibiting Glide SP docking scores ranging between −7.4 and −8.4 kcal/mol. Given that the Auto QSAR plot displayed some noise in this model, a decision was made to change the activity data to range from −6.00 and −8.9 kcal/mol for Auto QSAR _model_2 (see Appendix A). An improved regression coefficient was observed for Auto QSAR _model_3. It was noticed that the dynamic range was narrow and as expected, the correlation coefficient is poor. Therefore, a decision was made to add more molecules in the low-activity range of −6.0 kcal/mol. The next model was then able to predict the activity of the compounds in this range. As can be seen for Auto QSAR _model_4 up to Auto QSAR _model_7, and even with Auto QSAR _model _9, the same upward trend was observed (see Appendix A). The challenge now was to improve the ability of the models to predict the activities of the top-scoring compounds.

A plot showing activity (observed) vs. activity (predicted) is presented in Figure 3 for the most optimal active learning Auto QSAR models **8** and **10**. It is evident that Auto QSAR model_8 has some noisy data in the region between −6.5 kcal/mol and −9.5 kcal/mol. Therefore, it was then important to extend the range of the docking scores from −3.5 kcal/mol to −9.5 kcal/mol in Auto QSAR _model_10. The training data was improved with little noise in the region from −5 kcal/mol to −8.5 kcal/mol. Therefore, Auto QSAR _model_10 was able to predict the docking scores of the design ideas in the region between −5 kcal/mol and −8.5 kcal/mol, though it struggled to predict the activities of compounds with docking scores less than −5 kcal/mol. This is expected, as the model was not trained in the least active design ideas. Auto QSAR_model_10 was used to predict the docking scores of all the remaining compounds from all the rounds of enumerations. A total of 1252 poses were retrieved by docking the remaining design ideas against PL^pro^. Next, a core-constraint IFD docking approach was performed to filter compounds that do not meet the criteria. A total of 35 poses were selected with binding modes consistent with the native binding mode of co-crystalized GRL-0617 to SARS-CoV-2 PL^pro^ (PDB ID: 7JIR). This criterion further included selecting compounds with good docking scores, a good E-model score, and a good IFD score (Table 2), even though IFD did not manage to capture the binding mode of GRL-0617, our reference compound. Therefore, the compounds in Table 2 are ranked and ordered according to their docking scores. Each of the compounds generated a total of five poses and the top-scoring poses for each of the design ideas were selected and reported here.

The visual inspection of binding poses for decision-making purposes in drug discovery projects is very important [61]. Furthermore, the selection of the correct binding pose in docking is very important not only for decision-making purposes, but also for docking post-processing procedures such as MD simulations, and FEP+ simulations [46,51]. Therefore, in the literature, scientists have used procedures such as MD simulations employing binding pose metadynamics, [62] MD simulations, [14,63,64] and more recently, MCS docking, [50] including the most exhaustive and premium IFD-MD module [65], as docking post-processing procedures. This is aimed at modeling the correct native binding mode of a compound designed by De Novo to its prospective target.

Figure 4 details the distribution of docking scores to measure the activities of the design ideas that were prioritized for further relative binding affinity estimation using FEP+. Most of the design ideas exhibited docking scores ranging from −8 kcal/mol to −10 kcal/mol. Interestingly, even though the Auto QSAR models were trained on compounds with docking scores ranging from −3 to −10 kcal/mol, IFD was able to capture compounds with docking scores ranging from −10 to −12 kcal/mol (see Figure 4). Interestingly, the diverse set of enumerated ideas did not yield ideas that deviated from the initial physicochemical space and that had diminished potency (see Table 2). This demonstrated the applicability of the enumerated compounds to traverse chemical space and design ideas that are potent and can further be optimized against PL^pro^.

Concurring with this, Gentile et al. [66] proposed a deep-docking protocol that was used to dock an ultra-large library of drug-like molecules. The deep-docking scores were used to train and predict the potency of the subset of compounds from the database. Furthermore, Boyles et al. [67] performed a comparative study of training and testing a machine-learning model based on ligand poses from co-crystalized proteins against poses generated by docking scoring functions. The results revealed that the predictive power of machine learning models derived from docking poses had the same effect as poses generated from co-crystalized ligands in terms of their predictive ability of docking scores of the models [67]. Thus, this screening criteria interactively filters ideas with undesirable properties and low-scoring compounds. As a result, this approach is gaining traction in speeding up drug discovery efforts and allows for the rapid and accurate prediction of docking scores, which facilitates the prioritization of compounds for more thorough and computationally rigorous approaches like FEP+ prior to synthesis [54,60].

### 3.4. Relative Binding Affinity Prediction Using FEP+

Here, MDS was used to refine the PDB structure of 7JIR prior to FEP+ calculations. Since it has been reported in the literature that the bound conformation of a ligand sought from MDS improves the prediction of relative binding affinity by FEP+ [51]. As such, the 100 ns MDS frame was used to extract the bound conformation of GRL-0617 and was used as a reference compound to perform MCS docking and align the structures of the design ideas to the GRL-0617 MDS conformation. This was possible because the 100 ns MDS frame revealed that GRL-0617 was stable during the simulation window. Ligand fluctuations with respect to the receptor were below 3 Å (Figure 5a) during the 100 ns MDS trajectory. This indicates that the ligand did not move out of the active site pocket. Moreover, this further indicates that MDS was able to reproduce the native binding conformation of GRL-0617, which is similar to that exhibited by co-crystalized GRL-0617 to 7JIR. Again, the use of MCS docking has been reported to improve the relative binding affinities for FEP+ simulation [46].

Of the 35 compounds that were selected for FEP+ analysis, only 5 were calculated by FEP+, as some were not FEP+ amenable and others were not selected because of limited license tokens for FEP+. Therefore, after the ligand and protein health check was performed, the experimental IC_50_ value of GRL-0617 was converted to establish the extent of the change in free energy, ΔG. A blind 20 ns FEP+ simulation of the 4 design ideas that are FEP+ amenable was performed with GRL-0617 as a reference compound. FEP mapper was used to create a map with edges based on the ligand similarities (Appendix A). The MCS docking overlay of GRL-0617, and the design ideas demonstrate the structural activity relationship of the subset of compounds (Figure 5b) in the active site cavity of PL^pro^. The protein–ligand contacts between GRL-0617 and SARS-CoV-2 PL^pro^ reveal the native binding mode of GRL-0617 (Figure 5c,d). It should be noted that buried waters can decrease the binding energy, as the entropy of binding is penalized by the presence of buried waters, which should be displaced by the ligand (Figure 5d). A full-cycle closure technique was used for this 15 λ simulation window employing 20ns of MD simulation as per the procedure described in Section 2.7.

Figure 6a,b detail the plots of the free energy convergence and total change in free energy (ΔG in kcal/mol) between GRL-0617 (ligand 1) and compound **45** (ligand 2), as a function of time in the solvent and complex legs. The three plots for each leg also detail the reverse, forward, and sliding window with respect to the accumulated energy during the simulation window of the 20 ns trajectory. Therefore, the perturbation between the two ligands converged with a ΔG of −82.11 kcal/mol in the solvent leg and −81.63 kcal/mol in the complex leg. Further, the bootstrapping error estimates and analytical errors of 0.061 and 0.028 kcal/mol for the solvent leg and 0.108 and 0.028 kcal/mol for the complex leg, respectively, were achieved. Wang et al. [47], Cappel et al. [46], and Schindler et al. [68] have suggested that an accuracy of 1.4 kcal/mol is suitable in a drug discovery lead optimization stage. Concurring with this, Appendix A shows a full-circle closure perturbation map of design ideas and GRL-0617 detailing the difference in the binding free energy, ΔΔG (pink), between two ligands in an edge/node and their associated ligand similarity scores (green). As can be seen, GRL-0617 is 0.47 kcal/mol more active than compound **45**, with an error estimate of 0.88 kcal/mol, while GRL-0617 is 0.97 kcal/mol more active than compound **5**, with an error estimate of 1.40 kcal/mol.

On the other hand, GRL-0617 is 2.01 kcal/mol more active than compound **91**, with an error estimate of 0.99 kcal/mol. Furthermore, GRL-0617 is 4.64 kcal/mol more active than compound **23**, with an error estimate of 1.40 kcal/mol. On the other hand, compound **45** is 0.50 kcal/mol more active than compound **5**, with an error estimate of 0.88 kcal/mol, whilst compound **5** is 1.04 kcal/mol more active than compound **91**, with an error estimate of 1.40 kcal/mol. Lastly, compound **91** is 2.63 kcal/mol more active than compound **23**, with an error estimate of 1.40 kcal/mol (Appendix A).

Figure 6c,d, on the other hand, detail the histogram and ligand interaction diagram for the two ligands for the 15 endpoint λ replicas. This essentially demonstrates the percentage of the interaction between ligands 1 and 2, and PL^pro^. The ligand interaction diagram shows the hydrogen bond between the carbonyl group of GRL-0617 and Gln269, which occurs for 100% of the simulation window. A hydrogen bond between the NH group of GRL-0617 and Asp164 occurs for 50% of the simulation time, (Figure 6d) as is shown below, even though the MD simulation did not capture this hydrogen bonding between Asp164 and the NH group of GRL-0617 (Figure 5d). However, the binding mode is maintained between the active site cavity of PL^pro^ and GRL-0617. Furthermore, the π–π interaction network between 5-amino-2-methyl benzamide moiety and Tyr268, including the naphthalene moiety is shown. The ligand interaction diagram in Figure 6d also revealed the binding mode of compound **45** in the active site cavity of PL^pro^. There is a hydrogen bond between the carbonyl group in compound **45** and Gln269, which also occurs for 100% of the simulation time. There is a water-mediated hydrogen bond between the NH group of benzimidazole moiety and Gly266, which occurs for 41% of the simulation time. There is also another water-mediated hydrogen bond between the oxygen of the methyl sulfonyl moiety and Tyr268, which occurs for 21% of the simulation time. There is a further π–π interaction between Tyr268 and the benzene rings of 5-amino-2-methyl benzamide moiety and benzimidazole moieties.

The free energy convergence and total change in free energy (ΔG in kcal/mol) for compound **5** (ligand 1) and compound **91** (ligand 2) can be seen in Appendix A. Again, the perturbation between the two ligands converged with a ΔG of −58.81 kcal/mol in the solvent leg and −59.34 kcal/mol in the complex leg. Further, the bootstrapping error estimates and analytical errors of 0.050 and 0.039 for the solvent leg and 0.195 and 0.039 for the complex leg, respectively, were achieved. Appendix A show the histogram and ligand interaction diagram for the two ligands for the 15 endpoint λ replicas. The carbonyl group in the compound **5** hydrogen bonds with Gln269 for 100% of the simulation time and Tyr264 for 46% of the simulation time. Further, the hydrogen bond between the NH group of the benzamide moiety hydrogen bonds with Asp164 for 81% of the simulation time. The amino acid Asp164 and the Tyr273 hydrogen bond with the NH_2_ group in compound **5**, which occurs for 51% and 61% of the simulation time, respectively. A π–π interaction network between the aromatic ring of the benzamide moiety and Tyr264 including Tyr268 is observed.

Appendix A revealed that GRL-0617 exhibited a relative binding affinity estimate of −7.75 ± 0.40 kcal/mol which is consistent with the experimental binding affinity of −7.75 kcal/mol. Therefore, this explains the accuracy of FEP+ in reproducing the experimental binding affinity of this reference compound. Compound **45** in this series was the most active compound with a ΔG of −7.28 ± 0.96 kcal/mol. Compound **5** followed next with a ΔG of −6.78 ± 1.30 kcal/mol. The inactive compounds in this series were compound **91** and compound **23** with a ΔG of −5.74 ± 1.06 and −3.11 ± 1.45. In a drug discovery and development campaign, an error estimate of >1 kcal/mol is not acceptable [47,53,68]. However, the binding affinity error estimate can be improved by increasing the simulation window and using the MDS to model the system for starting structures prior to FEP+ simulations.

Appendix A shows buried waters in the node representation of GRL-0617 and in compound **45**’s binding poses predicted by FEP+, in the active site cavity of PL^pro^. We suggest the use of Water map to model the effect of active site waters on the enthalpy-entropy compensation. This will in turn, allow the determination of favorable and unfavorable waters that can be displaced by inserting functional groups that displace unfavorable waters to achieve a gain in potency [69]. Several researchers in the industry and in academia have used Water map for this purpose prior to FEP+ calculations with the aim of improving the relative binding affinities of the design ideas and repurposed drugs [70], an approach that we aim to follow for future studies in this research area. As such, the design of novel chemical matter containing 2-[2-(methanesulfonyl)propan-2-yl]-1*H*-benzimidazole, 1,3-dimethyl-1,3-dihydro-2*H*-benzimidazol-2-one, 3-[(*S*)-amino(4-fluorophenyl)methyl]-1,2,4-oxadiazol-5(4*H*)-one, and (*S*)-1-(4-methylphenyl)-1-(1*H*-pyrazol-3-yl)methanamine moieties, instead of the naphthalene moiety that is in GRL-0617, was successfully performed. Therefore, a combined strategy to demonstrate the synergy between active learning-based Glide docking tools in tandem with Pathfinder reaction-based enumeration tools and physics-based relative binding affinity estimation tools was successfully employed in this study.

## 4. Conclusions

The work reported here describes the use of Pathfinder-driven reaction-based enumeration to traverse chemical space by inserting building blocks and replacing the naphthalene moiety of GRL-0617, a lead compound, as an inhibitor of SARS-CoV-2 PL^pro^. This was followed by using PAINS, SMARTS, and physicochemical filters on the enumerated ideas. A subset of randomly selected ideas was placed into the active site cavity of PL^pro^ utilizing Glide SP, this was performed to filter ideas that do not bind with PL^pro^. Ideas with a docking score between −3 kcal/mol to −10.7 kcal/mol were selected to build the Auto QSAR models. The models were then used to predict the docking scores of the enumerated compounds. This process was repeated ten times, where the models were trained on compounds that were enumerated, tested, and re-scored. Interestingly, the models that were built and demonstrated good statistical results on average were models **6**, **7**, **8,** and **10**, with an R^2^ > 0.70, Q^2^ > 0.64, STDDEV < 0.30, and RMSE < 0.31. Furthermore, Auto QSAR model_10, was used to finally predict the docking scores of the candidate compounds from each of the rounds of enumerations. The top-scoring compounds were then selected for IFD calculations employing a core constraint. IFD generated a total of 1 252 poses and 35 of those poses were selected for FEP+ calculations based on visual inspection, IFD score, docking score, and E-model score.

MDS and MCS docking shed some light on the binding mode and structure–activity relationships of the design ideas through their alignment with the reference compound GRL-0617. Due to limitations with respect to license tokens for FEP+ webservices, we selected only 5 compounds to run the FEP+ calculations. The FEP+ results revealed that compound **45** and compound **5** were the most active compounds in the series with ΔG of −7.28 ± 0.96 kcal/mol, and −6.78 ± 1.30 kcal/mol. These binding affinities were due to strong hydrogen bonds, π–π interactions, and hydrophobic interactions, which were revealed by FEP+ analysis. The design ideas predicted here will be synthesized and tested in the future for their biological activities against wild-type PL^pro^ and viral cell lines containing various splice variations, and mutants of the SARS-CoV-2 spike protein. This will in turn help in preparing for future pandemics and the associated variants of SARS-CoV-2 and related viruses. Therefore, these combined strategies for lead optimization have great potential for exploring the unexplored regions of chemical space, with the hope of designing potent compounds with desirable physicochemical properties, which is important in drug discovery and development.

## Figures and Tables

**Figure 1 molecules-27-08569-f001:**
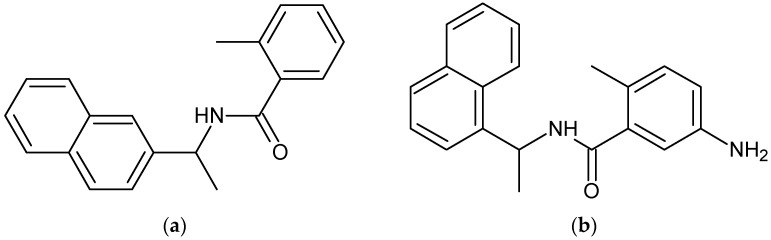
Structures of compound 7724772 (**a**) IUPAC name 2-methyl-*N*-[1-(naphthalen-2-yl) ethyl] benzamide and its optimized derivative GRL-0617 (**b**) with an IUPAC name 5-amino-2-methyl-*N*-[1-(naphthalen-1-yl) ethyl] benzamide.

**Figure 2 molecules-27-08569-f002:**
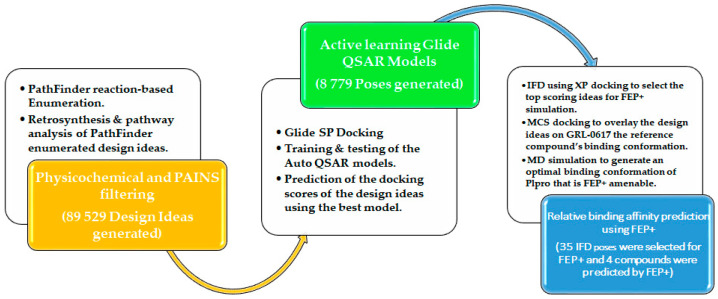
Pathfinder-driven reaction-based enumeration, active learning Glide QSAR modeling, IFD, and FEP+ screening workflow.

**Figure 3 molecules-27-08569-f003:**
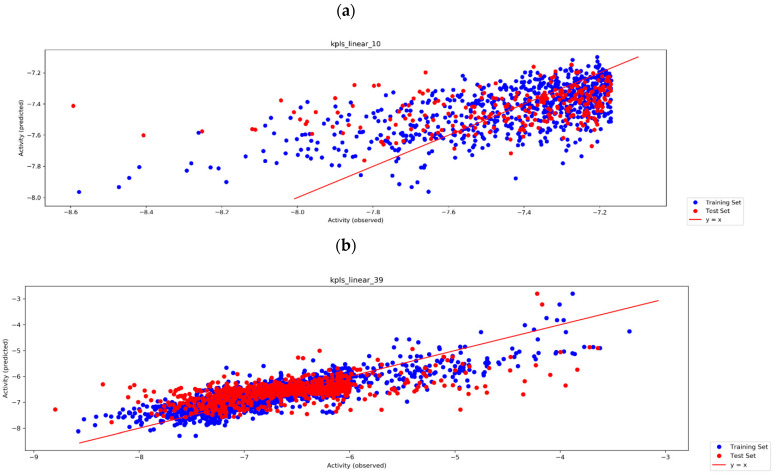
Auto QSAR active learning plots for model 8 (**a**) with an R^2^ of 0.7579 and a Q^2^ of 0.7191 and model 10 (**b**) with an R^2^ of 0.7410 and a Q^2^ of 0.3868.

**Figure 4 molecules-27-08569-f004:**
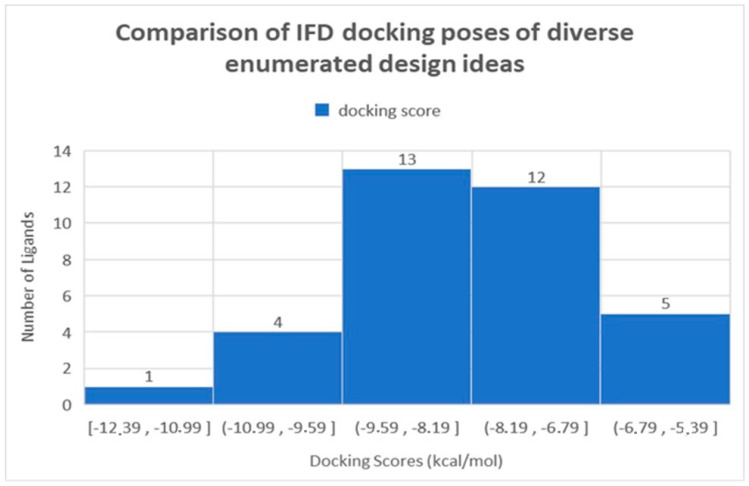
Histogram detailing the distribution of docking score (kcal/mol) for a diverse set of enumerated compounds measured using IFD, that were selected and prioritized for FEP+.

**Figure 5 molecules-27-08569-f005:**
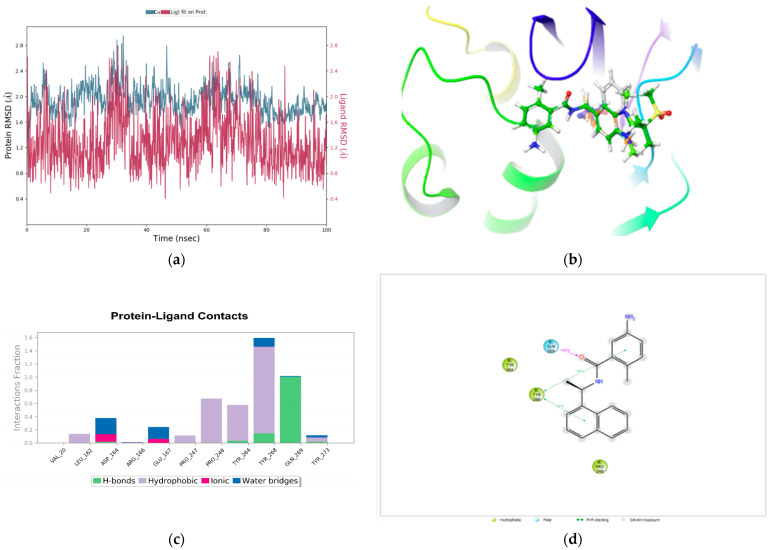
MD simulation of x-ray crystallography solved structure 7JIR and MCS docking overlay of the design ideas using the binding mode of GRL-0617 as a reference compound. (**a**) Root-mean-square deviation of GRL-0617 in the active site cavity of SARS-CoV-2 PL^pro^ obtained with an MD simulation time of 100 ns. (**b**) MCS docking overlay of design ideas selected prioritized and FEP+ amenable with GRL-0617 (grey carbon atoms) as a reference compound taken with the 100 ns MDS optimized frame. The carbon atoms of GRL-0617 are shown in grey, compound **45** in green, compound **91** in orange, compound **5** in faded green, and compound **23** in violet. The amino acids in the active site cavity were removed to provide a clear depiction of the overlay of the structures. (**c**) Histogram detailing the interaction between GRL-0617 and PL^pro^. (**d**) Ligand interaction diagram revealing hydrogen bonds, π–π, and hydrophobic interactions between GRL-0617 and PL^pro^.

**Figure 6 molecules-27-08569-f006:**
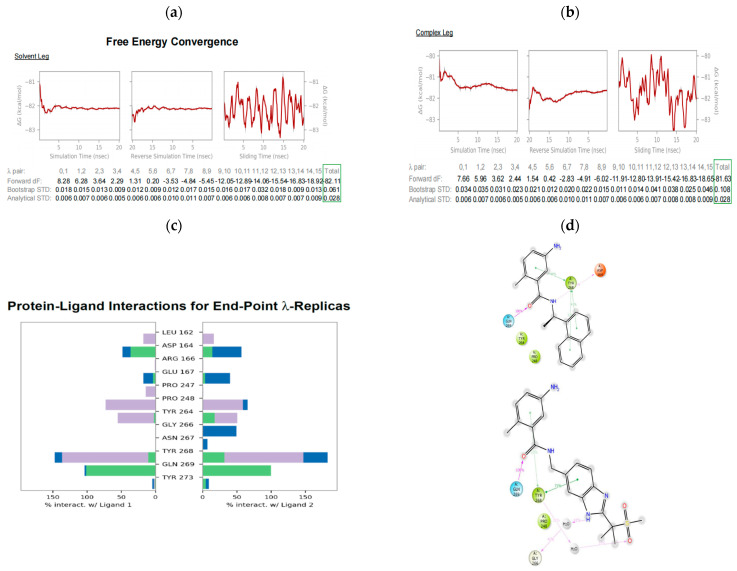
FEP+ simulation results for relative binding affinity prediction for the perturbation cycle of GRL-0617 (ligand 1) and compound **45** (ligand 2). (**a**) Free energy convergence of the perturbation cycle in solvent leg. (**b**) Free energy convergence of the perturbation cycle in complex leg. (**c**) Histogram depicting protein–ligand interactions for endpoint λ-replicas. (**d**) Ligand interaction diagram (Ligand 1 at the top and ligand 2 at the bottom) detailing the type of interactions observed between the two ligands and the receptor.

**Table 1 molecules-27-08569-t001:** The most optimum models generated by Auto QSAR to train the enumerated design ideas to predict their docking scores, following an active learning approach to assess their ability to bind to SARS-CoV-2 PL^pro^.

Model Name	Model Code	STDEV	RMSE	R^2^	Q^2^	#Factors	#Training Set	#Test Set
Auto QSAR _model_1	KPLS_linear_10	0.1805	0.2230	0.4310	0.1571	1	478	160
Auto QSAR _model_2	KPLS_radial_44	0.2261	0.2354	0.6262	0.5929	3	816	273
Auto QSAR _model_3	KPLS_radial_26	0.2284	0.2398	0.6090	0.5630	3	865	289
Auto QSAR _model_4	KPLS_radial_32	0.2474	0.2615	0.6227	0.5793	3	906	303
Auto QSAR _model_5	KPLS_molprint2D_8	0.2743	0.2983	0.6980	0.6450	3	1073	358
Auto QSAR _model_6	KPLS_molprint2D_48	0.2853	0.3040	0.7084	0.6676	3	1142	381
Auto QSAR _model_7	KPLS_molprint2D_47	0.3014	0.3091	0.7086	0.6910	3	1252	418
**Auto QSAR _model_8**	**KPLS_radial_42**	**0.2833**	**0.3053**	**0.7579**	**0.7191**	**4**	**1362**	**454**
Auto QSAR _model_9	KPLS_molprint2D_15	0.3323	0.3424	0.6718	0.6476	3	1662	554
**Auto QSAR_model_10**	**KPLS_linear_39**	**0.3444**	**0.5292**	**0.7410**	**0.3868**	**3**	**2391**	**797**

R^2^: Regression coefficient of the training set data; STDEV: Standard deviation; Q^2^: Regression quotient to determine the predictive ability of the test set; RMSE: Root-Mean-Square Error; # Factors: Number of multivariate factors.

**Table 2 molecules-27-08569-t002:** Physicochemical parameters simulated during model building of the design ideas including positive control inhibitors from the literature with their predicted docking scores obtained using Auto QSAR_model_10 and their associated IFD binding affinity parameters.

Rank Order No.	Enumeration Coupling Building Blocks/Pathway	Compound ID	TPSA	Num	Num	Mol	Mol	Pred Y	Pred Y	Docking	Glide	IFD	Exp IC_50_
				HDonors	HAcceptors	LogP	Wt		SD	Score	E-Model	Score	(uM)
1	N-Heterocycles	58	109.98	3	6	1.32	394.456	ND *	ND *	−12.39	−103.831	−664.209	
2	R^1^–R^2^_Carboxylates	39	110.52	3	5	3.60	425.529	−6.86	0.098	−10.93	−95.015	−664.167	
3	Pathway 10	1	135.08	3	8	2.64	546.481	−6.62	0.072	−10.87	−113.770	−673.770	
4	N-Heterocycles	128	83.8	3	3	3.01	320.396	ND *	ND *	−9.91	−78.744	−660.799	
5	N-Heterocycles	96	141.17	4	7	2.20	396.451	ND *	ND *	−9.64	−102.840	−660.007	
6	Carboxylates	5	82.05	2	5	1.26	310.357	−6.60	0.075	−9.44	−71.375	−660.109	
7	R^1^ (vinyl and aryl halides) R^2^ (Carboxylates)	2	82.05	2	5	1.26	310.357	−6.60	0.075	−9.44	−71.375	−660.109	
8	R^1^–R^2^_Carboxylates	36	123.13	3	7	3.02	449.845	−6.64	0.213	−9.12	−80.291	−659.220	
9	Carboxylates	1	75.43	2	4	2.63	309.369	−6.68	0.032	−8.81	−63.709	−658.183	
10	25 ^a^		ND *	ND *	ND *	ND *	ND *	ND *	ND *	−8.69	−76.159	−658.503	2.64
11	R^1^–R^2^_Carboxylates	49	120.56	3	7	2.61	388.431	−6.62	0.150	−8.55	−69.600	−653.725	
12	Pathway 10	24	99.42	2	4	5.13	448.906	−6.61	0.122	−8.53	−80.814	−658.802	
13	Primary and secondary amines	30	87.9	2	4	3.84	377.492	−7.60	0.090	−8.40	−76.477	−657.629	
14	vinyl and aryl halides_Trifluoroborates	24	53.71	1	4	4.53	403.469	−6.79	0.065	−8.39	−78.794	−658.165	
15	N-Heterocycles	67	103.77	4	3	3.34	372.428	ND *	ND *	−8.34	−90.436	−658.944	
16	R^1^ (vinyl and aryl halides) R^2^ (Carboxylates)	4	63.4	1	3	2.06	252.273	−6.76	0.054	−8.31	−53.957	−657.165	
17	Pathway 10	27	107.69	4	4	4.90	500.643	−6.79	0.091	−8.26	−97.525	−659.060	
18	2 ^a^		ND *	ND *	ND *	ND *	ND *	ND *	ND *	−8.25	−56.924	−656.513	8.70
19	24 ^a^		ND *	ND *	ND *	ND *	ND *	ND *	ND *	−8.14	−72.533	−658.903	0.56
20	N-Heterocycles	116	83.8	3	3	3.61	358.804	ND *	ND *	−8.10	−75.467	−658.695	
21	N-Heterocycles	57	70.91	3	2	4.49	377.822	ND *	ND *	−7.98	−76.563	−659.233	
22	N-Heterocycles	172	95.24	3	4	3.47	364.449	ND *	ND *	−7.94	−54.584	−656.907	
23	R^1^ (vinyl and aryl halides) R^2^ (Carboxylates)	1	89.34	3	4	0.72	273.13	−6.69	0.035	−7.77	−54.137	−660.284	
24	N-Heterocycles	91	83.8	3	3	3.13	320.396	ND *	ND *	−7.58	−87.783	−657.771	
25	1 ^a^		ND *	ND *	ND *	ND *	ND *	ND *	ND *	−7.48	−59.646	−655.252	200
26	N-Heterocycles	37	122.71	4	5	2.45	356.817	ND *	ND *	−7.46	−73.976	−657.600	
27	Primary and secondary amines	23	114.01	3	5	1.91	342.33	−7.55	0.077	−7.29	−60.112	−659.115	
28	Hydrazine-aryl	15	119.39	3	7	2.51	378,436	−7.55	0.068	−7.28	−66.860	−652.823	
29	Primary and secondary amines	24	80.9	2	4	3.81	360.461	−7.42	0.123	−7.25	−70.170	−657.073	
30	R^1^–R^2^_ Carboxylates	5	130.21	4	5	2.25	378.388	−6.62	0.090	−7.10	−59.593	−661.117	
31	Hydrazine-aryl	54	110.16	3	6	3.71	396.882	−7.55	0.068	−6.67	−57.155	−653.216	
32	vinyl and aryl halides_Trifluoroborates	11	92.51	3	7	3.94	425.558	−6.86	0.084	−6.52	−87.781	−658.149	
33	N-Heterocycles	45	117.94	3	5	2.66	400.504	ND *	ND *	−6.52	−74.218	−657.538	
34	N-Heterocycles	204	83.8	3	3	4.37	342.402	ND *	ND *	−6,56	−75.146	−655.866	
35	Hydrazine-aryl	35	110.16	3	6	2.95	380.427	−7.51	0.056	−6.51	−66.176	−653.539	

* ND—Not determined as the said compounds were part of the training set during Auto QSAR model building active learning process. ^a^ Compounds obtained from the literature with their experimental IC_50_ results [18,19]. TPSA—Total Polar Surface Area. LogP—Octanol/water partition coefficient. Mol Wt—Molecular Weight. Predicted Y—Predicted docking score (Y-intercept). PredY SD—Standard Deviation of the predicted docking score.

## Data Availability

Not applicable.

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
