# Peer review of "Pathfinder-Driven Chemical Space Exploration and Multiparameter Optimization in Tandem with Glide/IFD and QSAR-Based Active Learning Approach to Prioritize Design Ideas for FEP+ Calculations of SARS-CoV-2 PLpro Inhibitors"

_molecules, 2022, doi:10.3390/molecules27238569_

Round 1

Reviewer 1 Report

This is a very interesting and robust study that explores chemical space, with molecules filtered by AutoQSAR and docked using a recursive approach slowly building on the AutoQSAR model, using docking, MD and FEP+ to aid in the evaluation. At the end of this four compounds are identified to be excellent for further study for inhibition of the papain like protease of Covid-10.

Although this works well in supporting of the final conclusions, the presentation is complex and it becomes difficult to follow the stages of the study. For example looking at section 3.2 this could be shortened considerably and have a more meaningful impact on the reader with a figure  or flow diagram showing the appropriate chemistry and the numbers for each successive stage of filtering (perhaps with info also from Table 1). For identified systems it was difficult to follow in terms of relating the structures to the names in the text - this could me made much clearer.

This same type of presentation is difficult at other points, where in the conclusion for example GRL_0617 is compared one by one with each of the compounds - this needs to be concise to have the appropriate impact on the reader. In section 3.4 perhaps with attention to detail of individual systems, perhaps this presentation may be justified to provide the appropriate detail, in most cases however it dilutes the impact and detracts from the bigger picture of the study.

In terms of positive control inhibitors was there a criterion for inclusion - further discussion is possible?

Author Response

Thanks for reviewing the manuscript entitled: “Pathfinder-driven chemical space exploration and multiparameter optimization in tandem with Glide/IFD and QSAR-based active learning approach to prioritize design ideas for FEP+ calculations of SARS-COV-2 PLpro inhibitors”. I will address each of the reviewer’s comments in the following paragraphs below. My responses to each of the points will be in blue. Furthermore, in the manuscript all the changes made be it English language edits or the points addressing reviewer’s comments are written in red. While aspects that I felt one of the reviewers missed are highlighted in green. I hope that these responses have enough weight and substance to warrant publishing of this manuscript in your prestigious journal.

Reviewer 1

  1. Although this works well in supporting of the final conclusions, the presentation is complex, and it becomes difficult to follow the stages of the study. For example, looking at section 3.2 this could be shortened considerably and have a more meaningful impact on the reader with a figure or flow diagram showing the appropriate chemistry and the numbers for each successive stage of filtering (perhaps with info also from Table 1).

Response: A new figure 2 is added in section 3.1 with a process flow-diagram detailing the steps undertaken in this study. Please see line 221 – 224 in pg. 6 of the manuscript. Yes, the inclusion of figure 2 has significantly reduced the text in section 3.2 and 3.3 in the manuscript.

Incorporation of information from table 1 and 2 has been made in the text to aid in clearly explaining the chemistry. Please see line 272 and 280, pg. 8.  

  1. For identified systems it was difficult to follow in terms of relating the structures to the names in the text - this could me made much clearer.

Response: Appropriate chemical classes have been included in the explanation of the coupling pathways determined with retrosynthesis analysis. Further, information regarding the R-groups enumerated is given in line 215-2020, pg. 6. A correct flow of information is also detailed in figure 2 detailing the steps used for enumeration, Glide SP-based active learning Auto QSAR models, IFD screening, MDS, MCS docking prior to FEP+ calculations.  Please see line 221 – 224 in pg. 6 of the manuscript. A golden thread can be observed in Table 1, where the model names were corrected. Line 271 – 272, pg. 8. Also, the model names were also corrected in the text. Line 277-292, pg. 8 for example. Furthermore, in Table 2 the coupling building block’s chemical class and the compound ID were separated. This was done to be simpler to a reader and be able to follow the logic in this work. Please see line 388-389, pg. 12 and 13. Again, this correction in Table 2 has enabled a golden thread in how the names of the structures are reported in the text. See line 395, 407 and 408, pg. 13.

  1. This same type of presentation is difficult at other points, where in the conclusion for example GRL_0617 is compared one by one with each of the compounds - this needs to be concise to have the appropriate impact on the reader.

Response: This is corrected in the abstract and in the conclusion sections. See line 516-522, pg. 16.

  1. In section 3.4 perhaps with attention to detail of individual systems, perhaps this presentation may be justified to provide the appropriate detail, in most cases however it dilutes the impact and detracts from the bigger picture of the study.

Response: A correction to address this aspect is made in line 470-476, pg. 15. However, for the cycle closure/edge analysis the compounds connected from each edge are compared with the reference compounds in terms of their ΔΔG’s.

  1. In terms of positive control inhibitors was there a criterion for inclusion - further discussion is possible?

Response: The positive controls used were analogues of compound 7724772 and were designed together with GRL-0617 tested for bioanalysis using the same assay. They were included because they already have the IC50 values, so that they will be used to train, test, re-train, and re-test the QSAR models. Please see line 234-240, pg. 7.

Author Response

Reviewer 2

The followed methodology, model systems considered, and analysis methods are well

defined and establish a scientific rigor. This protocol could be adapted to other enzymes to find/design a new lead compounds. This article can be accepted and interest to wider drug community. Here are some minor format issues need to be considered.

  1. Table 1 and 2: Correct the numbers, there is a typo for all the numbers mentioned

here.

Response: See line 274-275 (Table 1) and line 391-392 (Table 2). 

  1. Figure 3: check the numbers of docking scores. There is a typo error.

Response: See line 326-327, pg. 10 with updated figure 3. However, the comma separating the numbers is used when generating the histogram plot and cannot be altered or edited in the figure. Since, the numbers when they are separated by the dot sign cannot generate the histogram plot for example. 

  1. Format the references and remove the hyperlinks, underlines.

Response:  Hyperlinks and underlines in the references were removed.

  1. Lines 596-599, check the text format.

Response: The font and font size are corrected. Line 480-482, pg. 16.

Reviewer 3 Report

The article entitled "Pathfinder-driven chemical space exploration and multiparameter optimization in tandem with Glide/IFD and QSAR-based active learning approach to prioritize design ideas for FEP+ calculations of SARS-COV-2 PLpro inhibitors" investigated GRL-0617 as potential PLpro inhibitor of SARS-CoV-2 virus. Although the article seems good-looking. however, the recent concern regarding SARS-CoV-2 is the variants for example omicron. Extensive mutations mainly in the spike protein made it more highly transmissible than other variants. I would like to suggest authors perform similar experiments with the omicron variant's main protein to compare their studied compounds effectiveness which would ultimately give a higher impact on the current work.

Closely related publications on the computational study should be discussed and referenced in the revised manuscript: PMID: 34739968, 29522307, 29300091, 35041375, 34329860, 34645849, 35301752, 33749528.

1. The introduction part is too long it must be precise.

2. Why the authors have chosen the 7JIR PDB structure though there are many updated PDB structures available with native ligands?

3. Why the authors have not used XP docking?

4. Did the authors have performed the re-docking of present ligands in the original PDB if present?

5. Ligand preparation has not been written. All the subtopics of the materials and methods section need major corrections.

6. Did the authors perform the energy minimization of all structures? If yes, please provide the details.

7. Please provide the formulas for all the required studies.

8. The conclusion is too chaotic.

9. MD simulation methods must be corrected.

10. The figures are poorly prepared and are not of publication quality.

11. The Structure figures should have an appropriate legend, in terms of domain, coloration, etc.

12. The labels in the 3D structures and shown in Arial font, whereas, other parts of the figures are in Calibri, I think font size, style should be consistent in all figures.

13. The problem again here is that the authors have really generated so much data it is not importantly useful to present such a study. I think the authors should have put effort into rationalizing the study, rather than just performing analysis after analysis.

Author Response

Reviewer 3

Although the article seems good-looking. however, the recent concern regarding SARS-CoV-2 is the variants for example omicron. Extensive mutations mainly in the spike protein made it more highly transmissible than other variants.

  1. I would like to suggest authors perform similar experiments with the omicron variant's main protein to compare their studied compounds effectiveness which would ultimately give a higher impact on the current work.

Response: Yes, I agree that the impact of this current work could be aided by looking at the binding of these compounds designed here on the wild type of SARS-COV-2 Spike protein, as well as the omicron mutants of SARS-COV-2 Spike protein. However, in this manuscript I have designed compounds to target SARS-COV-2 PLpro enzyme that is responsible for viral replication. Therefore, this approach cannot be possible computationally as there is currently no PDB structure of PLpro with omicron mutations. However, I have concluded in this study in line 524-526, that further synthesis and bioanalysis of the compounds designed in this study will be performed on vero cells containing wild type, and with cells infected with various mutants of SARS-COV-2 virus.   

  1. Closely related publications on the computational study should be discussed and referenced in the revised manuscript: PMID: 34739968, 29522307, 29300091, 35041375, 34329860, 34645849, 35301752, 33749528.

Response: I have managed to incorporate closely related articles to this work on SARS-COV-2 Mpro inhibitors by Arshia et al. [24], Murugesan et al. [25], and Patel et al. [26]. Please see line 69-74 of the introduction section.

  1. The introduction part is too long it must be precise.

Response: Yes, I agree. The introduction section is now significantly reduced from 3.5 pages to 1.5 pages.

  1. Why have the authors chosen the 7JIR PDB structure though there are many updated PDB structures available with native ligands?

Response: I have used various PDB structures of SARS-COV-2 PLpro in this study. I started with 7CJM, 7JIW, 7JN2 naming just a few to screen the compounds using the reported procedures. However, the problem starts during the receptor health check prior to FEP+ analysis. As you may already be aware that according to FEP+ best practices a receptor needs to pass a health check. Meaning that those that have incorrect conformations of the amino acid in the active site increases steric hindrance. Including improper torsions and the missing loops needed to be corrected with prime X. Even after minimizing these pdb structures with Prime X and corrected their conformers and included missing loops they did not pass the receptor health check and be FEP+ amenable. Therefore, 7JIR passed this receptor health check and hence all the simulations reported here were performed using the pdb structure.

  1. Why have the authors not used XP docking?

Response: XP docking was used in the IFD docking calculations to prioritise ideas for FEP+. Please see line 156-158, pg. 5 in section 2.5. Also, I have emphasized this in line 258 section 3.3, pg. 7.

  1. Did the authors have performed the re-docking of present ligands in the original PDB if present?

Response: The candidate compounds designed here are new chemical entities and have not yet been crystalized. It’s only GRL-0617 and some of its analogues that have pdb structures present. However, there is a new premium module from SchrÓ§dinger IFD-MD that can simulate the native binding conformation of a new chemical entity to its receptor. This then reduces the need to perform x-ray crystallography on the NCEs binding to the receptor. I plan to use this approach in future prior to FEP+ calculations.

  1. Ligand preparation has not been written. All the subtopics of the materials and methods section need major corrections.

Response: Materials and Methods section was revised including separating ligand preparation and protein preparation sections. See line 113-120, pg. 4.

  1. Did the authors perform the energy minimization of all structures? If yes, please provide the details.

Response: See line 119 on ligand preparation and line 129-130 in protein preparation section. Further, see Line 182-189 on Force-field builder prior to FEP+ analysis in section 2.8.

  1. Please provide the formulas for all the required studies.

Response: A new figure 2 is added in section 3.1 with a process flow-diagram detailing the steps undertaken in this study. Please see line 221 – 224 in pg. 6 of the manuscript. See Table 1 and table 2. See line 274-275 (Table 1) and line 391-392 (Table 2).  Including the appropriate in text inclusion. Appropriate chemical classes have been included in the explanation of the coupling pathways determined with retrosynthesis analysis. Further, information regarding the R-groups enumerated is given in line 215-2020, pg. 6. A correct flow of information is also detailed in figure 2 detailing the steps used for enumeration, Glide SP-based active learning Auto QSAR models, IFD screening, MDS, MCS docking prior to FEP+ calculations.  Please see line 221 – 224 in pg. 6 of the manuscript. A golden thread can be observed in Table 1, where the model names were corrected. Line 271 – 272, pg. 8. Also, the model names were also corrected in the text. Line 277-292, pg. 8 for example. Furthermore, in Table 2 the coupling building block’s chemical class and the compound ID were separated. This was done in order to be simpler to a reader to follow the logic in this work. Please see line 388-389, pg. 12 and 13. Again, this correction in Table 2 has enabled a golden thread in how the names of the structures are reported in the text. See line 395, 407 and 408, pg. 13.

Systematic names of the building blocks for the 4 analogues of GRL-0617 are given line 491-495.

  1. The conclusion is too chaotic.

Response: Conclusion has been refined.

  1. MD simulation methods must be corrected.

Response: See line 159-168, pg. 5 that details all the steps undertaken in MDS.

  1. The figures are poorly prepared and are not of publication quality.

Response: Figure 3 was updated to figure 4 and improved. See line 326-327, pg. 10.

Figure 4 cannot be updated as it is saved from the Auto QSAR panel directly. User cannot make any adjustments or any modifications.

Figure 5 a) has been updated. Figure 5d) legends were added. Line 414-417.

Figure 6 was improved including the contrast. Line 433-436.

  1. The Structure figures should have an appropriate legend, in terms of domain, coloration, etc.

Response: Figure 5b) has been updated colouration has been explained in the figure caption. Figure 5d) legends were added. Line 414-424.

  1. The labels in the 3D structures and shown in Arial font, whereas, other parts of the figures are in Calibri, I think font size, style should be consistent in all figures.

Response: Font size for the figure captions has been corrected. However, the font size for actual ligand interaction diagrams for MD simulation results (Fig. 5d) have a different font to that of FEP+ (Fig. 6d). These figures are generated as pdf files from their respective modules. Hence, the font type and font size cannot be altered.

  1. The problem again here is that the authors have really generated so much data it is not importantly useful to present such a study. I think the authors should have put effort into rationalizing the study, rather than just performing analysis after analysis.

Response: These steps as outlined here were necessary to screen large libraries of compounds generated by Pathfinder. Therefore, the Glide SP-based screening is similar to a single concentration high throughput screening experimentally. While FEP+ analysis is similar to a dose-response 10-point concentration experiment to determine the IC50 of candidate compounds. Therefore, all the other steps in between Glide IFD XP and FEP+ are important in that they are steps needed in FEP+’s best practice. They have been validated and verified by various academic groups and big pharma researchers.

Round 2

Reviewer 3 Report

Although the revised version of the manuscript improved a lot, however, some minor points are not satisfactorily answered. 

1. The image resolution must be corrected (Figure 2, Figure 4, Figure 5, Figure 6). In addition, Schrodinger software has a facility to export images in high resolution so take a look and provide high-resolution images so anyone can easily identify the residue interaction details.

2. There should be a uniform font size in the manuscript. (Table 1, Table 2 & Figure 4).

3. There are some typographical errors that must be rectified. For example, De Novo should be De novo.